# ARYABHATA: AN EXAM-FOCUSED LANGUAGE MODEL FOR JEE MATH

## ABSTRACT

We present **Aryabhata 1.0**, a compact 7B parameter math reasoning model optimized for the Indian academic exam, the Joint Entrance Examination (JEE). Despite rapid progress in large language models (LLMs), current models often remain unsuitable for educational use. Aryabhata 1.0 is built by merging strong open-weight reasoning models, followed by supervised fine-tuning (SFT) with curriculum learning on verified chain-of-thought (CoT) traces curated through best-of-$n$ rejection sampling. To further boost performance, we apply reinforcement learning with verifiable rewards (RLVR) using A2C objective with group-relative advantage estimation alongwith novel exploration strategies such as *Adaptive Group Resizing* and *Temperature Scaling*. Evaluated on both in-distribution (JEE Main 2025) and out-of-distribution (MATH, GSM8K) benchmarks, Aryabhata outperforms existing models in accuracy and efficiency, while offering pedagogically useful step-by-step reasoning. We release Aryabhata as a foundation model to advance exam-centric, open-source small language models..

## 1 INTRODUCTION

Large language models (LLMs) have shown remarkable progress in mathematical reasoning, but most existing systems fall short in supporting student learning in academic settings like India's Joint Entrance Examination (JEE). These exams require not only accurate solutions but also transparent and precise reasoning that aids student understanding and long-term learning.

We observe three broad classes of models in this space:

**Non-reasoning models** Instruction-tuned models (e.g., GPT-4o) were largely inaccurate on rigorous math exams like JEE. These models failed to perform multi-step reasoning, often guessing answers or relying on shallow pattern matching.

**Early reasoning models** introduced long chain-of-thought (CoT) reasoning to improve accuracy, with examples including OpenAI o1 (OpenAI, 2024) and DeepSeek R1 (DeepSeek-AI et al., 2025). While these models were more accurate than non-reasoning baselines, they remained impractical in real-world educational settings. For instance, o1 (OpenAI, 2024) did not expose its reasoning trace and provided just a summary of them, while DeepSeek R1 (DeepSeek-AI et al., 2025) produced long, nonlinear traces that made it difficult for students to follow the logic. Moreover, both models were relatively slow, generating lengthy explanations that consumed a significant amount of tokens and latency.

**Modern reasoning models** such as OpenAI o4-mini (OpenAI, 2025), Gemini 2.5 (Comanici et al., 2025), and the updated version of DeepSeek R1 (DeepSeek-AI et al., 2025) have improved further in raw accuracy and generation speed. However, pedagogical usability remains limited. For example, o4-mini (OpenAI, 2025) provides just a summary of its reasoning traces, while Gemini (Comanici et al., 2025) and DeepSeek R1 (DeepSeek-AI et al., 2025) still produce nonlinear, self-correcting reasoning paths that confuse learners rather than clarify concepts. (Samples are provided in Appendix D.)

In this work, we present **Aryabhata 1.0**, a compact and open 7B parameter model tailored for math reasoning in Indian competitive exams. Built via model merging and fine-tuned with domain-aligned data, Aryabhata combines accuracy, transparency, and efficiency, making it a viable foundation for educational AI applications.

## 2 RELATED WORK

Current math LLMs built on open-weight backbones have primarily leveraged Imitation Learning, Supervised Fine Tuning, and Reinforcement Learning to enhance chain-of-thought mathematical reasoning.

For instance **DeepSeekMath** (Shao et al., 2024), introduced in early 2024, advanced the capabilities of open weight models by pre-training on hundreds of billions of math-focused tokens and pioneering Group Relative Policy Optimization (GRPO).

**Qwen-2.5-Math-7B** (Yang et al., 2024) is a math-specialized 7B instruction-tuned model that supports chain-of-thought (CoT) and tool-integrated reasoning (TIR) across both English and Chinese problem sets.

NVIDIA's **AceMath-7B-Instruct** (Liu et al., 2025a), derived from Qwen, advances its performance further through a multi-stage SFT training pipeline designed to improve both mathematical and reasoning accuracy on multiple benchmarks and edging close to much larger models at 72B scale.

Meanwhile, **DeepSeek-R1** (DeepSeek-AI et al., 2025) introduced a pure RL-based reasoning model trained with GRPO-style verifiable rewards, achieving impressive results. Its distilled variants (**DeepSeek-R1-Distill-Qwen-7B** (DeepSeek-AI et al., 2025)) inherit reasoning performance via long CoT.

The **AceReason-Nemotron-7B** (Liu et al., 2025a) demonstrates that large-scale reinforcement learning can significantly enhance the reasoning capabilities of strong small- and mid-sized models by first training on math-only prompts, then on code-only prompts.

The **AceReason-Nemotron-1.1-7B** (Liu et al., 2025b) synergizes SFT and RL fine-tuning by employing a stage-wise RL approach on math-only and code-only prompts.

Our approach builds on these lines by merging models for hybrid capabilities (symbolic fluency + coherent CoT), followed by rejection-sampled SFT and RL with verifiable rewards, preserving both performance and efficiency in a compact model.

## 3 METHODOLOGY

The overall process can be categorized in the following four stages:

### 3.1 MODEL MERGING

The development of LLMs has seen a transition from System 1 (quick thinking) to System 2 (deliberate, methodical) reasoning, each with distinct advantages (Wu et al., 2025). While System 1 models excel at producing fluent answers with low latency, they often lack the depth required for complex reasoning. In contrast, System 2 models are capable of iterative self-correction and structured reasoning, but suffer from inefficiencies due to verbose or redundant CoT traces.

To address this challenge, Kimi k1.5 (Team et al., 2025) introduced the concept of merging reasoning and non-reasoning models, which was further explored by Wu et al. (2025). Building on this intuition, we carefully selected three distinct LLMs, each sharing the same base architecture.

- Qwen2.5-Math-7B-Instruct (Yang et al., 2024), a strong open source mathematical LLM providing solid baseline capabilities and fundamental math fluency.
- AceMath-7B-Instruct (Liu et al., 2025a) a version of Qwen 2.5 Math that was further fine-tuned by NVIDIA, significantly enhancing its accuracy on mathematical benchmarks.
- DeepSeek-R1-Distill-Qwen-7B (DeepSeek-AI et al., 2025), a long-form reasoning model derived by fine-tuning Qwen 2.5 Math on reasoning traces distilled from DeepSeek R1 (DeepSeek-AI et al., 2025).

We apply linear merging (Wortsman et al., 2022) to combine the models using the MergeKit (Goddard et al., 2024) framework.

Let $\theta_1$, $\theta_2$, $\theta_3$ be the parameters of Qwen, Ace, and DeepSeek, respectively. We compute:

$$\theta_{\text{merged}} = \alpha\theta_1 + \beta\theta_2 + \gamma\theta_3, \quad \text{where } \alpha + \beta + \gamma = 1$$

We select weights $\alpha, \beta, \gamma$ empirically based on the held-out math reasoning tasks. Final weights favor quickly addressing simpler problems while also performing methodical, multi-step analysis for more complex mathematical challenges.

Table 1: Topic-wise Question Distribution

| Topic | %age |
|---|---|
| Application of Derivatives | 4.50% |
| Application of Integrals | 2.27% |
| Binomial Theorem | 2.37% |
| Circles | 2.85% |
| Complex Numbers & Quadratic Equations | 6.00% |
| Conic Section | 7.55% |
| Continuity and Differentiability | 2.71% |
| Definite Integration | 2.45% |
| Determinants | 3.04% |
| Differential Equations | 3.77% |
| Indefinite Integration | 3.26% |
| Inverse Trigonometric Functions | 5.31% |
| Limits and Derivatives | 3.88% |
| Matrices | 2.46% |
| Permutations and Combinations | 4.23% |
| Probability | 5.69% |
| Quadratic Equations | 4.45% |
| Relations and Functions | 2.24% |
| Sequence and Series | 2.75% |
| Sets | 1.04% |
| Statistics | 1.89% |
| Straight Lines | 2.31% |
| Three Dimensional Geometry | 3.92% |
| Trigonometric Functions | 4.51% |
| Vector Algebra | 2.89% |
| Miscellaneous | 11.65% |

## 3.2 DATA CURATION

High-quality, domain-aligned data is essential for training effective mathematical reasoning models. To this end, we relied on a proprietary corpus curated by out subject matter experts and educators, ensuring close alignment with the Indian Joint Entrance Examination (JEE) standards. This dataset represents years of academic effort and is considered our core intellectual property. As such, we do not publicly release the training data.

We parsed approximately 250,000 raw questions from internal databases. To ensure syntactic coherence and semantic relevance, we applied the following filtering steps:

- Removed diagram-based questions, which require multimodal reasoning not supported by current text-only models.

- Filtered out non-English or poorly formatted questions.

- Stripped all answer options from the remaining questions to frame the task as open-ended generation rather than classification. This design choice was also explored by Chandak et al. (2025)

- Since we stripped options from the questions, we removed the questions which relied on options to be answered such as "which of the following is true"

Table 2: Chain-of-Thought generation outcomes from best-of-4 sampling.

| Correct CoTs | # Questions | Total CoTs | Usage |
|:---:|:---:|:---:|:---:|
| 0 | 31,470 | 0 | Used in RLVR only |
| 1 | 9,647 | 9,647 | SFT |
| 2 | 9,066 | 18,132 | SFT |
| 3 | 12,643 | 37,929 | SFT |
| 4 | 67,247 | 268,988 | 10% sampled for SFT |

To standardize and clean raw question-answer pairs, we designed a structured prompt (see Appendix A) that extracts the core question, normalizes the answer format, identifies dependencies and detects the question language, using OpenAI o4-mini.

This process resulted in a clean dataset of around 130,000 questions suitable for the generation of further chain of thought. The topic-wise distribution of questions is outlined in Table 1.

### 3.3 SUPERVISED FINE-TUNING WITH REJECTION SAMPLING

To generate high-quality chain-of-thought (CoT) supervision, we employed best-of-4 rejection sampling using the merged model. For each curated question $x$, we sampled four CoT responses $\{y_1, y_2, y_3, y_4\}$, and selected only those completions whose final answer matched the known correct answer i.e. $GT(x)$, using Algorithm 4. This filtering process ensures logical correctness and minimizes noisy supervision signals.

We then grouped the questions based on how many of the four generations lead to the correct answers and selected samples for curriculum-style supervised fine-tuning (Bengio et al., 2009), i.e., beginning the training with easier samples (e.g., 4/4 correct) and gradually introducing harder examples (e.g., 3/4, 2/4, 1/4 correct). This curriculum-based training stabilizes early learning and improves generalization on harder problems.

Let $\mathcal{D}_{\text{sft}} = \{(x^{(i)}, y^{(i)})\}_{i=1}^N$ denote the dataset of input questions and their corresponding verified CoT completions. The supervised fine-tuning objective minimizes the standard next-token prediction loss:

$$\mathcal{L}_{\text{SFT}} = - \sum_{(x,y) \in \mathcal{D}_{\text{sft}}} \sum_{t=1}^{T} \log(p_\theta(y_t \mid x, y_{<t})) \tag{1}$$

where $y_t$ is the $t^{\text{th}}$ token of the target CoT sequence, and $p_\theta$ is the model's probability distribution parameterized by $\theta$.

In total, we obtained approximately 350,000 verified CoTs across around 100,000 questions, which were sampled to serve as the core training corpus for SFT, as detailed in Table 2. The 0/4 cases were retained for downstream reinforcement learning with verifiable rewards (RLVR) to further improve coverage and robustness in challenging problem spaces.

We used Parameter Efficient Finetuning, particulary Low-Rank Adaptation (Hu et al., 2021) during SFT using peft (Mangrulkar et al., 2022) library, the training parameters are mentioned in Appendix C.

### 3.4 REINFORCEMENT LEARNING WITH VERIFIABLE REWARDS

We extend Reinforcement Learning with Verifiable Rewards (RLVR) (Lambert et al., 2025) by incorporating group-based advantage estimation (Shao et al., 2024) within an Advantage Actor-Critic (A2C) framework (Mnih et al., 2016) .

#### 3.4.1 GROUP-RELATIVE POLICY OPTIMIZATION

Our approach optimizes the following A2C objective with group-relative advantage estimation:

$$J^{A2C}(\theta) = \mathbb{E}(\alpha_i) \sim \pi\theta \left[ \frac{1}{G} \sum_{i=1}^{G} \frac{1}{|\alpha_i|} \log \pi_\theta(\alpha_i) \cdot \tilde{A}_i \right]$$

We optimize the A2C objective over G sampled response sequences $\alpha_i$, applying length-normalized gradients weighted by sequence-level advantages $\tilde{A}_i$ computed through group-relative advantage estimation.

**Binary Reward Structure**: We employ a simple binary reward that provides unambiguous feedback for mathematical reasoning:

$$R_i = \begin{cases} 1 & \text{if the final answer is correct} \\ 0 & \text{if the final answer is incorrect} \end{cases}$$

**Group Advantage Estimation** The advantage function is computed using group-relative normalization:

$\hat{A}_{i,t} = \frac{R_i - \bar{R}_{\text{group}}}{\sigma_{\text{group}}}$

where $\bar{R}_{\text{group}}$ is the mean reward across all solutions in the group and $\sigma_{\text{group}}$ is the standard deviation.

**Key Benefits**: This group-relative baseline offers several advantages:

- **Reduced variance**: Group comparison stabilizes gradient estimates
- **Simplified training**: Eliminates need for KL divergence constraints or probability ratio clipping
- **Natural compatibility**: Works seamlessly with binary rewards, common in mathematical reasoning tasks

### 3.4.2 EXPLORATION STRATEGIES

**Adaptive Group Sizing**: Unlike fixed group sizes in standard GRPO implementations (von Werra et al. (2020), Sheng et al. (2024), Daniel Han & team (2023)), we dynamically adjust group size based on problem difficulty. Starting with a group size of 8 for simpler problems, we scale up to a group size of 64 for harder ones.

The dynamic group size follows:

$$G_d = 8 \times 2^k$$

where $k \in \{0, 1, 2, 3\}$ is determined by the group average reward $\bar{R}_{\text{group}}$. When performance drops below preset thresholds, we increase $k$, scaling groups as: $8 \to 16 \to 32 \to 64$.

This adaptive scaling improves sampling diversity and advantage estimation stability for challenging problems while efficiently allocating computational resources.

**Progressive Temperature Scaling**: We continuously increase the sampling temperature from 0.6 to 1.0 throughout training, this was explored in contemporary works like POLARIS (An et al., 2025). This progressive scaling balances exploitation and exploration:

- **Initial phase**: Low temperature (0.6) promotes training stability through conservative sampling
- **Progressive increase**: Temperature gradually rises, encouraging more diverse solution exploration
- **Final phase**: Temperature reaches 1.0, enabling much more exploration of the action space compared to lower temperatures.

**Curriculum-Based Sampling**: We filter training samples to focus on an optimal difficulty range, removing both trivial and intractable problems:

- **Too easy**: Provide minimal learning signal due to high success rates
- **Too hard**: Introduce noise through consistently low performance

Our filtering uses a difficulty assessment function $f_{\text{difficulty}}(x)$ based on model success rates:

$$\mathcal{D}_t^{\text{filtered}} = \{x \in \mathcal{D}_t : \alpha_{\min} \leq f_{\text{difficulty}}(x) \leq \alpha_{\max}\}$$

This curriculum approach concentrates computational resources on problems that maximize learning progress.

### 3.4.3 HARDWARE-OPTIMIZED ALTERNATING INFERENCE-TRAINING PIPELINE

To maximize computational resource utilization and overcome GPU memory constraints inherent in large-scale reinforcement learning training, we implement an alternating synchronized rollout strategy that decouples the inference and training phases into discrete, non-overlapping computational cycles.

Our approach leverages vLLM (Kwon et al., 2023) as the primary inference engine for rollout generation. The training pipeline operates according to the following synchronized cycle:

**Phase 1: Rollout Generation**

- vLLM inference engine is created
- Batch rollout generation is performed across all training samples
- Generated rollouts are serialized and stored in system memory
- vLLM process is destroyed and releases all GPU memory allocations

**Phase 2: Policy Optimization**

- Training model is loaded onto GPU memory with full memory availability
- Policy gradient updates are computed using stored rollouts
- Model parameters are updated and checkpointed
- Training model is offloaded to memory
- vLLM process is restarted for subsequent rollout generation

This alternating architecture provides several critical advantages: **(1) Memory Efficiency**: Complete GPU memory is available to each phase, enabling larger batch sizes and model configurations than would be possible with concurrent inference-training approaches. **(2) Training Stability**: Deterministic separation of inference and training phases eliminates potential race conditions and memory fragmentation issues.

### 3.4.4 TRAINING CONFIGURATION AND HYPERPARAMETERS

Our reinforcement learning implementation employs carefully tuned hyperparameters optimized for mathematical reasoning tasks while maintaining computational efficiency within hardware constraints. The training configuration incorporates modern optimization techniques and memory-efficient strategies to enable stable convergence at scale.

**Optimization Configuration**: We utilize the Adam optimizer (Kingma & Ba, 2017) with a conservative learning rate of $1 \times 10^{-6}$ to ensure stable policy gradient updates throughout the training process.

**Memory and Precision Management**: Training is conducted using bfloat16 (BF16) mixed precision arithmetic, which provides substantial memory savings while maintaining numerical stability for gradient computations. Gradient checkpointing is employed to further reduce memory consumption during backpropagation, enabling training of larger models within available GPU memory constraints.

**Sequence and Batch Configuration**: The model operates within a maximum context length of 4,096 tokens, providing sufficient capacity for complex multi-step mathematical reasoning while maintaining computational tractability.

# 4 EVALUATION

We evaluated Aryabhata 1.0 across both in-distribution and out-of-distribution math benchmarks to assess its accuracy and efficiency in solving problems at scale.

We evaluate model-generated solutions using the `pass@1` accuracy. The solutions are generated using greedy decoding (temperature = 0). To determine whether a predicted answer matches the ground-truth answer for a question, we follow the pipeline described in the Algorithm 4.

Answer Matching Procedure [1] **Input:** Predicted answer $a_p$, Ground-truth answer $a_g$, Options (if any) **Output:** Match status (True / False)

$a_p = a_g$ or sympy_latex_match($a_p$, $a_g$) True option/identifier from $a_p$ == option/identifier from $a_g$ True Query LLM judge with $a_p$, $a_g$, and options (if any) LLM returns YES True False

Depending on whether the question is Multiple Choice Question or a Numerical Answer Type, we use different prompts to query the judge model (GPT-4o-mini). The prompts are provided in Table 4.

## 4.1 IN-DISTRIBUTION EVALUATION: JEE MAIN 2025

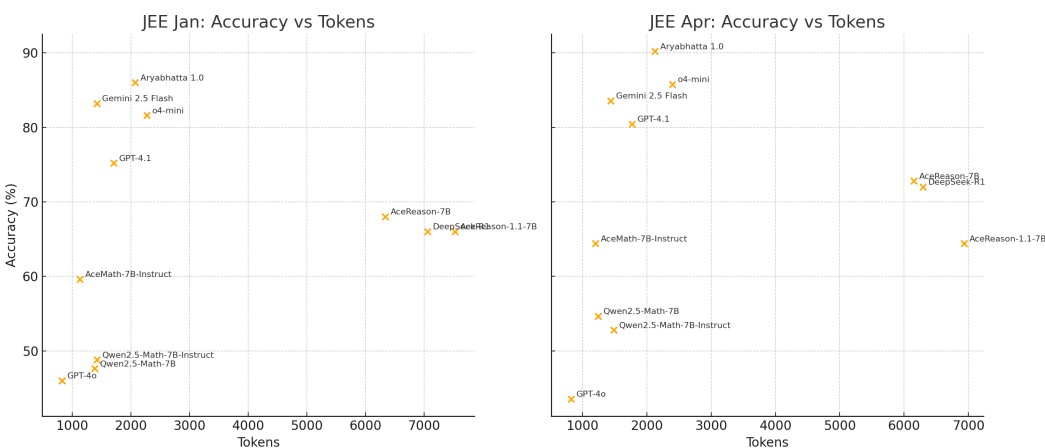

Figure 1: Scatter plots showing Accuracy vs. Tokens for JEE Jan and JEE Apr.

Table 3: Performance comparison on MATH 500 and GSM8K benchmarks

| Model | MATH 500 | GSM8K |
|---|---|---|
| Aryabhatta 1.0 | 83.6 | 94.8 |
| Qwen/Qwen2.5-Math-7B-Instruct | 66.0 | 94.7 |
| nvidia/AceMath-7B-Instruct | 80.6 | 93.4 |
| GPT-4o | 69.2 | 94.6 |
| deepseek-ai/DeepSeek-R1-Distill-Qwen-7B | 85.2 | 69.7 |
| nvidia/AceReason-Nemotron-7B | 84.2 | 76.5 |
| nvidia/AceReason-Nemotron-1.1-7B | 85.4 | 93.1 |
| GPT-4.1 | 86.6 | 94.0 |
| o4-mini | 94.8 | 90.1 |
| Gemini 2.5 Flash | 93.6 | 85.1 |

To measure performance in familiar distribution settings, we evaluate Aryabhata on the JEE Main 2025 exam. The January session contains 250 questions (10 papers with 25 questions each), while

the April session comprises 225 questions (9 papers with 25 questions each), all sourced from official exam papers.

Figure 1 shows that Aryabhata 1.0 achieves an accuracy of **86.0%** on the January session and **90.2%** on the April session, while maintaining token efficiency with an average of approximately ˜2K tokens per response.

Compared to both open-weight and proprietary models, Aryabhata outperforms all baselines in accuracy while remaining competitive in inference cost.

## 4.2    OUT-OF-DISTRIBUTION EVALUATION

To evaluate generalization beyond the fine-tuning distribution, we benchmark Aryabhata 1.0 on the following datasets:

- **MATH 500**: A curated benchmark of 500 competition-style problems drawn from the larger MATH dataset originally introduced by Hendrycks et al. (2021).
- **GSM8K** (Cobbe et al., 2021): A widely used benchmark of grade school math word problems.

Table 3 shows that Aryabhata demonstrates **competitive generalization** to unseen tasks of comparable difficulty, outperforming its base models on both MATH and GSM8K.

## CONCLUSION AND FUTURE WORK

In this work, we introduced **Aryabhata 1.0**, a compact open source model with 7B parameters for mathematical reasoning, specifically designed for the Indian competitive exam ecosystem. By merging diverse mathematical LLMs and fine-tuning on carefully curated and verified domain-specific data, Aryabhata achieves state-of-the-art performance on in-distribution benchmarks such as JEE Main, while demonstrating competitive generalization to out-of-distribution tasks like MATH and GSM8K.

Looking ahead, we plan to: Expand coverage to Physics and Chemistry, building similar reasoning capabilities in other STEM domains. Scale to the full syllabus across Foundation, JEE (Main & Advanced), and NEET, enabling end-to-end subject-level assistance. Develop a family of exam-centric, open source small language models (SLMs) that are compact, efficient, and aligned to Indian education standards.

We believe that this direction will empower millions of students with accessible and curriculum-aligned AI tools that complement classroom learning and personalized preparation.

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

## A  PROMPT FOR QUESTION CLEANING

Listing 1: Prompt used for Question Cleaning

```
Clean and standardize math questions by removing multiple-choice
options, normalizing the answer format, identifying dependencies,
and determining the language. For any answers expressed in
MathML, convert them to LaTeX. Conversion of MathML in the
**question** is *not required* (but preserve LaTeX if already
present).
Additionally, provide a clear **step-by-step reasoning**
explaining how each part of the output was derived.

### Instructions:

1. Identify and extract the core question text:
    * Remove all multiple-choice options (e.g., A-D or 1 4 ),
    ensuring the main question remains grammatically and
    semantically intact.
    * Preserve existing LaTeX in the question.
    * Do **not** convert MathML in the question. It may be
    retained as-is.

2. Normalize the answer:
    * If the answer is given as an option label (e.g., "Answer:
    B"), replace it with the corresponding value from the
    provided options.
    * If the answer is already a value, retain it.
    * If the answer is in MathML, convert it to LaTeX.

3. Determine dependency flags:
    * **Option-dependent:** Is the question understandable and
    solvable without access to the answer options? Mark `True` if
    the question lacks key information without them; otherwise,
    `False`.
    * **Diagram-dependent:** Does the question reference or rely
    on a diagram, figure, or visual element? Mark `True` or
    `False`.

4. Identify the language:
    * Detect and report the language of the question text (e.g.,
    `English`, `Hindi`, `Tamil`, etc.).

5. Provide reasoning:
    * For each output field (question, answer, flags, language),
    include a clear explanation of how the output was determined.
    * The reasoning should follow a logical step-by-step format,
    but does **not** need to be wrapped in any special `<reason>`
    block.

# Output Format

<question> cleaned question </question>
<answer> cleaned answer </answer>
<option_dependent> True/False </option_dependent>
<diagram_dependent> True/False </diagram_dependent>
<language> detected language </language>

* All math in the **answer** must be in LaTeX.
```

```
* There should be **no references** to original option labels
(e.g., "A", "1", or "Option B").
* Ensure the cleaned question is coherent, self-contained, and
grammatically  correct.
* The reasoning can be in free-text form and must explain how
each part of the output  was derived.

### Example 1
Input:
What is the derivative of \(x^2 + 3x + 5\)?
A) \(2x + 3\)
B) \(x + 3\)
C) \(x^2 + 3\)
D) \(2x + 5\)
Answer: A

Output:
<question> What is the derivative of \(x^2 + 3x + 5\)? </question>
<answer> \(2x + 3\) </answer>
<option_dependent> False </option_dependent>
<diagram_dependent> False </diagram_dependent>
<language> English </language>
\end{verbatim}

\begin{verbatim}
### Example 2
Input:
<p>Simplify the following expression:</p>
<math xmlns="http://www.w3.org/1998/Math/MathML">
  <mfrac>
    <msqrt>
      <msup><mi>a</mi><mn>2</mn></msup>
    </msqrt>
    <mi>a</mi>
  </mfrac>
</math>

<p>Options:</p>
1) <math
xmlns="http://www.w3.org/1998/Math/MathML"><msqrt><mi>a</mi>
</msqrt></math>
2) <math
xmlns="http://www.w3.org/1998/Math/MathML"><mi>a</mi></math>
3) <math
xmlns="http://www.w3.org/1998/Math/MathML"><mfrac><mn>1</mn><mi>a
</mi></mfrac></math>
4) <math
xmlns="http://www.w3.org/1998/Math/MathML"><mn>1</mn></math>

Answer: 1

Output:
<question> Simplify the following expression:
<math xmlns="http://www.w3.org/1998/Math/MathML">
  <mfrac>
    <msqrt>
      <msup><mi>a</mi><mn>2</mn></msup>
    </msqrt>
    <mi>a</mi>
```

```
    </mfrac>
</math>
</question>
<answer> \sqrt{a} </answer>
<option_dependent> False </option_dependent>
<diagram_dependent> False </diagram_dependent>
<language> English </language>
```

## B  PROMPTS FOR ANSWER MATCHING

| MCQ | Numerical |
|---|---|
| **System Prompt:**

You are checking an MCQ. Given the list of options, determine if answer 1 and answer 2 are the same. Answer 1 is the same as answer 2 only if all the options match. Reason step-by-step and put the final answer YES or NO in \boxed{}. | **System Prompt:**

You are checking an exam. For a given question, determine if answer 1 and answer 2 are the same. Since the answers are for the same question, you can assume similar  context for both answers and make appropriate assumptions when checking if they are the same. Reason step-by-step and put the  final answerYES or NO in \boxed{}. |
| **User Prompt:**

Options:
A: <Option 1>
B: <Option 2>
C: <Option 3>
D: <Option 4>
answer 1: <Correct Answer>
answer 2: <Predicted Answer> | **User Prompt:**

answer 1: <Correct Answer>
answer 2: <Predicted Answer> |

Table 4: Prompts used for Answer Matching

## C  HYPER-PARAMETERS FOR SUPERVISED FINE TUNING

The hyper-parameters for LoRA are provided in the Table 5 and the hyper-parameters for SFT are provided in the Table 6.

Table 5: PEFT configuration using LoRA.

| Parameter | Value |
| --- | --- |
| Rank | 128 |
| LoRA Alpha | 128 |
| LoRA Dropout | 0.1 |
| Bias | none |
| Target Modules | {q_proj, k_proj, v_proj, o_proj, gate_proj, up_proj, down_proj, embeddings} |

Table 6: Training configuration used for supervised fine-tuning.

| Parameter | Value |
| --- | --- |
| Precision | bfloat16 |
| Max Sequence Length | 16,384 |
| Batch Size (per device) | 1 |
| Gradient Accumulation Steps | 16 |
| Effective Batch Size | 16 |
| Number of Epochs | 3 |
| Initial Learning Rate | $2 \times 10^{-5}$ |
| Final Learning Rate | $2 \times 10^{-7}$ |
| Learning Rate Scheduler | Linear |
| Optimizer | AdamW (8-bit) |
| Warmup Steps | 5 |
| Packing | False |
| Logging Steps | 1 |
| WandB Reporting | Enabled |

## D  EXAMPLE MODEL RESPONSES

The sample question along with its correct answer is presented in Figure 2.
The response generated by GPT-4o is shown in Figure 3.
The response produced by DeepSeek R1 Distill Qwen 7B is illustrated across Figures 4, 5, and 6.
The response from Aryabhata 1.0 is depicted in Figure 7.

$Question$ :

Let $y = f(x)$ be the solution of the differential equation $\frac{dy}{dx} + \frac{xy}{x^2-1} = \frac{x^4+4x}{\sqrt{1-x^2}}, -1 < x < 1$ such that $f(0) = 0$. If $6 \int_{-1/2}^{1/2} f(x)dx = 2\pi - \alpha$ then $\alpha^2$ is equal to:

Answer:

27

Figure 2: Sample question with the correct answer

Given the differential equation and initial condition:

$$\frac{dy}{dx} + \frac{xy}{x^2-1} = \frac{x^4+4x}{\sqrt{1-x^2}}, \quad -1 < x < 1 \quad \text{with} \quad f(0) = 0,$$

we solve the differential equation using the method of integrating factors. The integrating factor, $\mu(x)$, is given by:

$$\mu(x) = e^{\int \frac{x}{x^2-1}\,dx}.$$

To find this, we simplify the integral:

$$\int \frac{x}{x^2-1}\,dx.$$

We use partial fractions:

$$\frac{x}{x^2-1} = \frac{x}{(x-1)(x+1)} = \frac{1/2}{x-1} + \frac{1/2}{x+1}.$$

Thus,

$$\int \frac{x}{x^2-1}\,dx = \int \frac{1/2}{x-1}\,dx + \int \frac{1/2}{x+1}\,dx = \frac{1}{2}\ln|x-1| + \frac{1}{2}\ln|x+1| = \frac{1}{2}\ln|(x-1)(x+1)|.$$

So, the integrating factor is:

$$\mu(x) = e^{\frac{1}{2}\ln|(x-1)(x+1)|} = \sqrt{|(x-1)(x+1)|}.$$

Since $-1 < x < 1$, $\sqrt{|(x-1)(x+1)|}$ simplifies to $\sqrt{1-x^2}$.

Multiplying the differential equation by the integrating factor:

$$\sqrt{1-x^2}\frac{dy}{dx} + \frac{x\sqrt{1-x^2}y}{x^2-1} = x^4 + 4x.$$

This rearranges to:

$$\frac{d}{dx}(y\sqrt{1-x^2}) = x^4 + 4x.$$

Integrating both sides w.r.t. $x$:

$$y\sqrt{1-x^2} = \int (x^4 + 4x)\,dx.$$

Let:

$$\int x^4\,dx = \frac{x^5}{5}, \quad \int 4x\,dx = 2x^2.$$

Thus:

$$y\sqrt{1-x^2} = \frac{x^5}{5} + 2x^2 + C.$$

Given $f(0) = 0$:

$$0 = \frac{0^5}{5} + 2\cdot 0^2 + C \implies C = 0.$$

So:

$$y\sqrt{1-x^2} = \frac{x^5}{5} + 2x^2.$$

And:

$$y = \frac{x^5/5 + 2x^2}{\sqrt{1-x^2}}.$$

We want $\int_{-1/2}^{1/2} f(x)\,dx = \int_{-1/2}^{1/2} \frac{x^5/5+2x^2}{\sqrt{1-x^2}}\,dx$.

Solving using symmetry and standard integral forms: Thus:

$$\int_{-1/2}^{1/2} f(x)dx = \frac{2\pi}{6} - \frac{\alpha}{6}.$$

Hence, equating both expressions:

$$6\int_{-1/2}^{1/2} f(x)dx = 2\pi - \alpha.$$

Thus:

$$\alpha = \frac{\pi}{3},$$

and: $\alpha^2 = \left(\frac{\pi}{3}\right)^2 = \frac{\pi^2}{9}$. So: $\boxed{\dfrac{\pi^2}{9}}$.

Figure 3: Response from GPT-4o (Part 1 of 1)

1350
1351
1352
1353
1354
1355
1356
1357
1358
1359
1360
1361
1362
1363
1364
1365
1366
1367
1368
1369
1370
1371
1372
1373
1374
1375
1376
1377
1378
1379
1380
1381
1382
1383
1384
1385
1386
1387
1388
1389
1390
1391
1392
1393
1394
1395
1396
1397
1398
1399
1400
1401
1402
1403

<think> Okay, so I have this differential equation to solve: $\frac{dy}{dx} + \frac{xy}{x^2-1} = \frac{x^4+4x}{\sqrt{1-x^2}}$, with the condition that $f(0) = 0$. Then, after finding $f(x)$, I need to compute $6 \int_{-1/2}^{1/2} f(x)\, dx$, which equals $2\pi - \alpha$. Finally, I have to find $\alpha^2$. Hmm, that seems a bit involved, but let's break it down step by step.

First, the differential equation is linear. It's in the form $\frac{dy}{dx} + P(x)y = Q(x)$. So, I can use an integrating factor to solve it. The standard method for linear differential equations is to find an integrating factor $\mu(x) = \exp\left(\int P(x)\, dx\right)$. Then, multiply both sides of the equation by $\mu(x)$ and integrate.

Let me write down the equation again:

$$\frac{dy}{dx} + \left(\frac{x}{x^2-1}\right)y = \frac{x^4+4x}{\sqrt{1-x^2}}$$

So, $P(x) = \frac{x}{x^2-1}$, and $Q(x) = \frac{x^4+4x}{\sqrt{1-x^2}}$.

First, let's compute the integrating factor $\mu(x)$:

$$\mu(x) = \exp\left(\int \frac{x}{x^2-1}\, dx\right)$$

Hmm, the integral of $\frac{x}{x^2-1}\, dx$. Let me make a substitution. Let $u = x^2 - 1$, then $du/dx = 2x$, so $\frac{1}{2}du = x\, dx$. Therefore, the integral becomes $\frac{1}{2}\int \frac{du}{u} = \frac{1}{2}\ln|u| + C = \frac{1}{2}\ln|x^2 - 1| + C$.

So,

$$\mu(x) = \exp\left(\frac{1}{2}\ln|x^2-1|\right) = |x^2-1|^{1/2}$$

Since we're working in the interval $-1 < x < 1$, $x^2 - 1$ is negative, so $|x^2 - 1| = 1 - x^2$. Therefore, $\mu(x) = \sqrt{1-x^2}$.

Wait, that's interesting. So, the integrating factor is $\sqrt{1-x^2}$. Let me write that down:

$$\mu(x) = \sqrt{1-x^2}$$

Now, multiply both sides of the differential equation by $\mu(x)$:

$$\sqrt{1-x^2}\frac{dy}{dx} + \sqrt{1-x^2}\cdot\left(\frac{x}{x^2-1}\right)y = \frac{(x^4+4x)}{\sqrt{1-x^2}}\cdot\sqrt{1-x^2}$$

Simplify each term:
First term: $\sqrt{1-x^2}\frac{dy}{dx}$
Second term: $\sqrt{1-x^2}\cdot\left(\frac{x}{x^2-1}\right)y = \frac{x}{\sqrt{1-x^2}(x^2-1)}y$
Wait, $x^2 - 1$ is negative, so $x^2 - 1 = -(1 - x^2)$. Therefore,

$$\frac{x}{\sqrt{1-x^2}(x^2-1)} = \frac{-x}{(1-x^2)^{3/2}}$$

Wait, maybe I should think differently. Let me compute the left-hand side after multiplying by $\mu(x)$. The left-hand side should be the derivative of $\mu(x)y$. Let me check:

$$\frac{d}{dx}[\mu(x)y] = \mu'(x)y + \mu(x)\frac{dy}{dx}$$

So, let's compute $\mu'(x)$:

$$\mu(x) = (1-x^2)^{1/2}, \quad \mu'(x) = \frac{1}{2}(1-x^2)^{-1/2}(-2x) = \frac{-x}{\sqrt{1-x^2}}$$

Therefore,

$$\frac{d}{dx}[\mu(x)y] = \frac{-x}{\sqrt{1-x^2}}y + \sqrt{1-x^2}\frac{dy}{dx}$$

Which is exactly the left-hand side of the equation after multiplying by $\mu(x)$. So, that's correct.
Therefore, the equation becomes:

$$\frac{d}{dx}[\sqrt{1-x^2}y] = x^4 + 4x$$

Now, to solve for $y$, we can integrate both sides with respect to $x$:

$$\sqrt{1-x^2}y = \int(x^4+4x)\, dx + C$$

Compute the integral on the right:

$$\int(x^4+4x)dx = \int x^4 dx + \int 4x dx = \frac{x^5}{5} + 2x^2 + C$$

So,

$$\sqrt{1-x^2}y = \frac{x^5}{5} + 2x^2 + C$$

Figure 4: Response from DeepSeek R1 Distill Qwen 7B (Part 1 of 3)

Therefore, solving for $y$:

$$y = \frac{\frac{x^5}{5} + 2x^2 + C}{\sqrt{1 - x^2}}$$

So, that's the general solution. Now, apply the initial condition $f(0) = 0$. Let's plug $x = 0$ into the solution:

$$y(0) = \frac{0 + 0 + C}{1} = C$$

But $y(0) = 0$, so $C = 0$.
Therefore, the particular solution is:

$$y = \frac{\frac{x^5}{5} + 2x^2}{\sqrt{1 - x^2}} = \frac{x^5 + 10x^2}{5\sqrt{1 - x^2}}$$

So, $f(x) = \frac{x^5 + 10x^2}{5\sqrt{1-x^2}}$
Now, the next step is to compute

$$6 \int_{-1/2}^{1/2} f(x)\, dx = 2\pi - \alpha$$

Given:

$$f(x) = \frac{x^5 + 10x^2}{5\sqrt{1 - x^2}} = \frac{x^5}{5\sqrt{1 - x^2}} + \frac{2x^2}{\sqrt{1 - x^2}}$$

So the integral becomes:

$$\int_{-1/2}^{1/2} \left( \frac{x^5}{5\sqrt{1 - x^2}} + \frac{2x^2}{\sqrt{1 - x^2}} \right) dx$$

Split into two integrals:

$$\frac{1}{5} \int_{-1/2}^{1/2} \frac{x^5}{\sqrt{1 - x^2}} dx + 2 \int_{-1/2}^{1/2} \frac{x^2}{\sqrt{1 - x^2}} dx$$

Note: The first integrand is an odd function (since $x^5$ is odd), and the limits are symmetric, so the integral is zero.
So we compute:

$$2 \int_{-1/2}^{1/2} \frac{x^2}{\sqrt{1 - x^2}} dx = 4 \int_0^{1/2} \frac{x^2}{\sqrt{1 - x^2}} dx$$

Use the substitution $x = \sin\theta$, $dx = \cos\theta\, d\theta$, $\sqrt{1 - x^2} = \cos\theta$. When $x = 0$, $\theta = 0$; when $x = \frac{1}{2}$, $\theta = \frac{\pi}{6}$
So the integral becomes:

$$\int_0^{\pi/6} \sin^2\theta\, d\theta = \frac{1}{2} \int_0^{\pi/6} (1 - \cos 2\theta)\, d\theta = \frac{1}{2} \left[ \theta - \frac{\sin 2\theta}{2} \right]_0^{\pi/6}$$

$$= \frac{1}{2} \left( \frac{\pi}{6} - \frac{\sin(\pi/3)}{2} \right) = \frac{1}{2} \left( \frac{\pi}{6} - \frac{\sqrt{3}}{4} \right) = \frac{\pi}{12} - \frac{\sqrt{3}}{8}$$

Then,

$$\int_{-1/2}^{1/2} f(x)\, dx = \frac{\pi}{3} - \frac{\sqrt{3}}{2}$$

Now,

$$6 \int_{-1/2}^{1/2} f(x)\, dx = 6 \left( \frac{\pi}{3} - \frac{\sqrt{3}}{2} \right) = 2\pi - 3\sqrt{3}$$

Given $6 \int_{-1/2}^{1/2} f(x)\, dx = 2\pi - \alpha$, we conclude:

$$\alpha = 3\sqrt{3} \Rightarrow \alpha^2 = 27$$

$$\boxed{27}$$

</think>
Given the differential equation:

$$\frac{dy}{dx} + \frac{xy}{x^2 - 1} = \frac{x^4 + 4x}{\sqrt{1 - x^2}}, \quad -1 < x < 1$$

with the initial condition $f(0) = 0$, we need to find $\alpha^2$ where $6 \int_{-1/2}^{1/2} f(x)\, dx = 2\pi - \alpha$.

Figure 5: Response from DeepSeek R1 Distill Qwen 7B (Part 2 of 3)

First, we solve the differential equation using the integrating factor method. The integrating factor is:

$$\mu(x) = \exp\left(\int \frac{x}{x^2 - 1}\, dx\right) = \sqrt{1 - x^2}$$

Multiplying through by the integrating factor, we get:

$$\sqrt{1 - x^2}\frac{dy}{dx} + \frac{x}{\sqrt{1 - x^2}}y = x^4 + 4x$$

This simplifies to:

$$\frac{d}{dx}\left(\sqrt{1 - x^2}y\right) = x^4 + 4x$$

Integrating both sides:

$$\sqrt{1 - x^2}y = \int (x^4 + 4x)\, dx = \frac{x^5}{5} + 2x^2 + C$$

Applying the initial condition $f(0) = 0$, we find $C = 0$. Thus, the solution is:

$$y = \frac{\frac{x^5}{5} + 2x^2}{\sqrt{1 - x^2}} = \frac{x^5 + 10x^2}{5\sqrt{1 - x^2}}$$

Next, we compute the integral:

$$6\int_{-1/2}^{1/2} f(x)\, dx$$

We split the integral into two parts:

$$6\left(\int_{-1/2}^{1/2} \frac{x^5}{5\sqrt{1 - x^2}}\, dx + \int_{-1/2}^{1/2} \frac{2x^2}{\sqrt{1 - x^2}}\, dx\right)$$

The first integral is zero because it is an odd function. The second integral is computed using the substitution $x = \sin\theta$:

$$4\int_0^{\pi/6} \sin^2\theta\, d\theta = 4\left(\frac{\pi}{12} - \frac{\sqrt{3}}{8}\right) = \frac{\pi}{3} - \frac{\sqrt{3}}{2}$$

Thus,

$$6\left(\frac{\pi}{3} - \frac{\sqrt{3}}{2}\right) = 2\pi - 3\sqrt{3}$$

Given $6\int_{-1/2}^{1/2} f(x)\, dx = 2\pi - \alpha$, we find $\alpha = 3\sqrt{3}$. Therefore, $\alpha^2 = (3\sqrt{3})^2 = 27$.

$$\boxed{27}$$

Figure 6: Response from DeepSeek R1 Distill Qwen 7B (Part 3 of 3)

To solve the given differential equation $\frac{dy}{dx} + \frac{xy}{x^2-1} = \frac{x^4+4x}{\sqrt{1-x^2}}$ with the initial condition $f(0) = 0$, we start by identifying it as a first-order linear differential equation. The standard form of a first-order linear differential equation is $\frac{dy}{dx} + P(x)y = Q(x)$, where $P(x) = \frac{x}{x^2-1}$ and $Q(x) = \frac{x^4+4x}{\sqrt{1-x^2}}$.

The integrating factor $\mu(x)$ is given by $\mu(x) = e^{\int P(x)\,dx} = e^{\int \frac{x}{x^2-1}\,dx}$. To find this integral, we use the substitution $u = x^2 - 1$, so $du = 2x\,dx$. Thus, $\int \frac{x}{x^2-1}\,dx = \frac{1}{2} \int \frac{1}{u}\,du = \frac{1}{2} \ln|u| = \frac{1}{2} \ln|x^2 - 1|$. Therefore, the integrating factor is $\mu(x) = e^{\frac{1}{2}\ln|x^2-1|} = \sqrt{|x^2 - 1|} = \sqrt{1 - x^2}$ since $x$ is in the interval $(-1, 1)$.

Multiplying both sides of the differential equation by the integrating factor $\sqrt{1-x^2}$, we get:

$$\sqrt{1-x^2}\frac{dy}{dx} + \frac{x}{\sqrt{1-x^2}}y = x^4 + 4x.$$

The left-hand side is the derivative of $y\sqrt{1-x^2}$:

$$\frac{d}{dx}\left(y\sqrt{1-x^2}\right) = x^4 + 4x.$$

Integrating both sides with respect to $x$, we have:

$$y\sqrt{1-x^2} = \int (x^4 + 4x)\,dx.$$

The integral on the right-hand side is:

$$\int x^4\,dx + \int 4x\,dx = \frac{x^5}{5} + 2x^2 + C.$$

Thus,

$$y\sqrt{1-x^2} = \frac{x^5}{5} + 2x^2 + C.$$

Solving for $y$, we get:

$$y = \frac{\frac{x^5}{5} + 2x^2 + C}{\sqrt{1-x^2}}.$$

Using the initial condition $f(0) = 0$, we substitute $x = 0$ and $y = 0$:

$$0 = \frac{0 + 0 + C}{\sqrt{1-0^2}} = C.$$

So, $C = 0$, and the solution is:

$$y = \frac{x^5 + 10x^2}{5\sqrt{1-x^2}}.$$

We need to find $6\int_{-1/2}^{1/2} f(x)\,dx$. Since $f(x) = \frac{x^5+10x^2}{5\sqrt{1-x^2}}$, we have:

$$6\int_{-1/2}^{1/2} \frac{x^5 + 10x^2}{5\sqrt{1-x^2}}\,dx = \frac{6}{5}\int_{-1/2}^{1/2} \frac{x^5 + 10x^2}{\sqrt{1-x^2}}\,dx.$$

We can split this integral into two parts:

$$\frac{6}{5}\left(\int_{-1/2}^{1/2} \frac{x^5}{\sqrt{1-x^2}}\,dx + 10\int_{-1/2}^{1/2} \frac{x^2}{\sqrt{1-x^2}}\,dx\right).$$

The first integral is zero because $\frac{x^5}{\sqrt{1-x^2}}$ is an odd function integrated over a symmetric interval. So, we only need to evaluate the second integral:

$$\frac{6}{5}\cdot 10 \int_{-1/2}^{1/2} \frac{x^2}{\sqrt{1-x^2}}\,dx = 12\int_{-1/2}^{1/2} \frac{x^2}{\sqrt{1-x^2}}\,dx.$$

Since $\frac{x^2}{\sqrt{1-x^2}}$ is an even function, we can rewrite the integral as:

$$12\cdot 2 \int_{0}^{1/2} \frac{x^2}{\sqrt{1-x^2}}\,dx = 24\int_{0}^{1/2} \frac{x^2}{\sqrt{1-x^2}}\,dx.$$

To evaluate this integral, we use the substitution $x = \sin\theta$, so $dx = \cos\theta\,d\theta$ and when $x = 0$, $\theta = 0$, and when $x = 1/2$, $\theta = \frac{\pi}{6}$. The integral becomes:

$$24\int_{0}^{\pi/6} \frac{\sin^2\theta}{\sqrt{1-\sin^2\theta}}\cos\theta\,d\theta = 24\int_{0}^{\pi/6} \frac{\sin^2\theta}{\cos\theta}\cos\theta\,d\theta = 24\int_{0}^{\pi/6} \sin^2\theta\,d\theta.$$

Using the identity $\sin^2\theta = \frac{1-\cos 2\theta}{2}$, we get:

$$24\int_{0}^{\pi/6} \frac{1-\cos 2\theta}{2}\,d\theta = 12\int_{0}^{\pi/6} (1-\cos 2\theta)\,d\theta = 12\left[\theta - \frac{\sin 2\theta}{2}\right]_{0}^{\pi/6}.$$

Evaluating this, we have:

$$12\left(\frac{\pi}{6} - \frac{\sin\frac{\pi}{3}}{2} - 0 + 0\right) = 12\left(\frac{\pi}{6} - \frac{\sqrt{3}}{4}\right) = 2\pi - 3\sqrt{3}.$$

We are given that $6\int_{-1/2}^{1/2} f(x)\,dx = 2\pi - \alpha$, so $2\pi - 3\sqrt{3} = 2\pi - \alpha$. Therefore, $\alpha = 3\sqrt{3}$, and $\alpha^2 = (3\sqrt{3})^2 = 27$.

The final answer is:

$$\boxed{27}.$$

Figure 7: Response from Aryabhata 1.0 (Part 1 of 1)