# OpenReview forum: "Aryabhata: An exam-focused language model for JEE Math"
_ICLR.cc/2026/Conference — Submitted to ICLR 2026_

### Official Review · Reviewer_x2XF · 2025-10-27

**Soundness:** 1
**Presentation:** 1
**Contribution:** 1
**Rating:** 0
**Confidence:** 4

**Summary:**

This paper tries various existing model technologies and combines them into a training solution for JEE, then test their model on JEE Main 2025, MATH500 and GSM8K.

**Strengths:**

A model was trained for Joint Entrance Examination (JEE), maybe useful for Indian academic exam.

**Weaknesses:**

This is more like an experiment report using a variety of techniques, and the combination of these techniques is not informative.
Lacks ablation experiments, for example, does not fuse model weights but uses a single model, and the necessity of the SFT stage.
The dataset used for model training is not publicly available, making it unclear whether the results are reproducible.
During the evaluation phase, verification was performed only on GSM8K and MATH500. The benchmark itself is not difficult, and no general applicability testing was performed.

**Questions:**

Q1: Why not use open source data to verify the method? It is impossible to know whether the improvement in model performance comes from the data itself rather than the training method.

Q2: What is the innovation of this paper?

---

### Official Review · Reviewer_BitB · 2025-10-30

**Soundness:** 2
**Presentation:** 3
**Contribution:** 2
**Rating:** 2
**Confidence:** 4

**Summary:**

This work introduces Aryabhata 1.0, a compact 7B parameter math reasoning model optimized for the Indian academic exam. The method consists of merging strong open-weight reasoning models, SFT and RLVR. The final model outperforms existing models in accuracy and efficiency.

**Strengths:**

1. The paper is well-written and easy to follow.

2. It presents an engineering-oriented approach to training a reasoning model for India’s Joint Entrance Examination (JEE), combining model merging, supervised fine-tuning, and reinforcement learning in a clear and organized manner.

**Weaknesses:**

1. The proposed methodology is mainly engineering-oriented, without addressing a clear research challenge or presenting substantial methodological novelty. The techniques described, such as group-relative advantage estimation and exploration strategies like adaptive group resizing and temperature scaling, have been explored in prior works and appear incremental.

2. More importantly, the paper lacks ablation studies to validate the effectiveness of the proposed components. Without such analysis, it is difficult to assess the contribution of each part of the training pipeline and the overall necessity of the proposed approach.

**Questions:**

See the weaknesses section

---

### Official Review · Reviewer_4sGY · 2025-10-31

**Soundness:** 2
**Presentation:** 1
**Contribution:** 1
**Rating:** 2
**Confidence:** 5

**Summary:**

This paper proposed Aryabhata, a compact 7B-parameter language model specifically designed to solve JEE math problems. This work used a multi-stage pipeline: merging three specialized 7B models; curating a 130K-question JEE-aligned dataset; fine-tuning via sft, reject sampling and reinforcement learning. The model achieves state-of-the-art performance on JEE 2025.

**Strengths:**

1. End-to-end. This paper covers various mainstream technologies, from model merging, to data acquisition, and then to different training methods such as SFT and RL.

**Weaknesses:**

1. This paper does not address any scientific questions. Although it employs a variety of techniques, it reads more like a technical report, and the scientific motivation for using LLMs to solve JEE problems is not clearly articulated.
2. The paper lacks baseline methods. It does not present the performance after model merging or the results at each training stage, making it unclear which techniques are actually effective.
3. The papepr does not conduct ablation studies on the methods used. The proposed 'Adaptive Group Sizing' method lacks experimental evidence to demonstrate its effectiveness.
4. GSM8K and MATH cannot serve as out-of-domain benchmarks. They are too simple for today's models.
5. Poor writing quality and presentation: The overall writing is very poor, including formatting, length. and it is Verbose and Unfocused.

**Questions:**

1. Can you provide more clear motivation for solving JEE problems?
2. The authors need to provide more ablation results to show the effectness of proposed methods.
3. What is the key factor to obtain the sota performance of JEE problems?

---

### Official Review · Reviewer_gZxq · 2025-11-01

**Soundness:** 2
**Presentation:** 1
**Contribution:** 1
**Rating:** 2
**Confidence:** 4

**Summary:**

This paper presents **Aryabhata 1.0**, a 7B-parameter small language model (SLM) specifically designed for mathematical reasoning in India’s high-stakes Joint Entrance Examination (JEE).
The authors train the model through a sequential pipeline involving **Model Merging**, **Supervised Fine-Tuning (SFT)**, and **Reinforcement Learning with Verifiable Rewards (RLVR)**.
On the MATH500 benchmark, Aryabhata 1.0 outperforms several open-source 7B models.

**Strengths:**

1. The paper adopts a standard fine-tuning (post-training) pipeline, which is a reasonable approach for developing a domain-expert model tailored to a specific application scenario.
2. The model demonstrates superior performance on the JEE benchmark compared to other existing models.

**Weaknesses:**

1. The paper does not introduce any new methods or insights; it merely combines existing techniques.
2. The paper lacks comprehensive ablation studies to analyze the contribution of each component. For instance, it remains unclear how much of the **performance gain comes from** Model Merging, SFT, or RLVR. Questions such as how the model would perform without Model Merging or without SFT are left unanswered.
3. The paper does not evaluate the model beyond mathematics, despite claiming that “we release Aryabhata as a foundation model to advance exam-centric applications.” In practice, small models often suffer from catastrophic forgetting after domain-specific fine-tuning. The authors are encouraged to test Aryabhata on broader benchmarks such as MMLU-Pro to verify its general capabilities.
4. The model’s performance on MATH500 is not the best; for example, it does not surpass DeepSeek-R1-Distill-Qwen-7B. Moreover, GSM8K is too simple for higher-education-level evaluation, and its score has limited significance.
5. The paper suffers from presentation issues: Table 1 is overly large for its limited importance, Figure 1 has text that is too small, and the authors’ own model is not highlighted in bold for easy comparison.

**Questions:**

I am particularly curious about the second weakness, i.e., the lack of ablation studies. Specifically:

1. What is the model’s performance after each of the three stages: Model Merging, SFT, and RLVR?
2. How would the final performance degrade if any of these stages were skipped?

---

### Meta-Review · Area_Chair_Ndvt · 2026-01-05

**Summary:**

This paper presents a math reasoning language model specifically optimized for the Indian Joint Entrance Examination.  Unfortunately, the reviewers unanimously recommended rejecting the paper.  The major concern was that the work doesn't present any novel methods or convey useful knowledge / experiments for the community and acts more as a technical report describing the engineering of the system.  Multiple reviewers were concerned that the lack of any ablations for a pipeline of steps makes it very difficult for others to understand where any performance gains come from.  The authors don't seem to have provided a rebuttal.  Therefore the recommendation is to reject the paper.

**Reviewer Concerns:**

There is no rebuttal.

**Reviewer Scores:**

No rebuttal, and therefore it seems unlikely anyone would have changed their scores.

---

### Decision · Program_Chairs · 2026-01-26

Reject